# Position: Express Your Doubts – Probabilistic World Modeling Should Not Be Based on Token *logprobs*

**Eitan Wagner** [1]  **Omri Abend** [1]

## Abstract

Language modeling has shifted in recent years from a distribution over strings to prediction models with textual inputs and outputs for general-purpose tasks. This position paper highlights the often overlooked implications of this shift for the use of large language models (LLMs) as probability estimators, especially for world probabilities. In light of the theoretical distinction between distribution estimation and response prediction, we examine LLM training phases and common use cases for LLM output probabilities. We show that the different settings lead to distinct, potentially conflicting, desired output distributions. This lack of clarity leads to pitfalls when using output probabilities as event probabilities. Our position advocates for second-order prediction—incorporating probabilities explicitly as part of the output—as a theoretically sound method, in contrast to using token logprobs. We conclude with suggestions for potential directions to improve the probabilistic soundness of this method.

## 1  Introduction

Probabilistic reasoning about the world is deeply rooted in artificial intelligence (Russell & Norvig, 1995; Pearl, 2009) and cognitive science (Oaksford & Chater, 2007). As a primary source of human knowledge acquisition is communication (Tomasello, 2009), the abundance of textual data available on the Internet plays a significant role. While the theoretical ability to derive world knowledge based on language alone sparks ongoing debates (Searle, 1980; Bender & Koller, 2020; Pinker, 2007; Piantadosi & Hill, 2022), in practice, modern large language models (LLMs) show

impressive performance on world-related tasks, even when trained exclusively on text (Brown et al., 2020).

**Language modeling** originally referred to a probability distribution over finite strings of tokens from a given vocabulary (Shannon, 1948; Bahl et al., 1983). With the rise of LLMs, the common use has shifted (Rogers & Luccioni, 2024). The change occurs along two dimensions: (1) The learning task has shifted from **distribution estimation**—learning to match the language's data-generating distribution—to **response prediction**, outputting "correct" responses to queries (Chung et al., 2022; Brown et al., 2020). (2) Learning has shifted from the source (consisting of language or "words") to a target which consists of facts and events (or "world"). Despite these changes, models still typically output probabilities over tokens (obtained as the softmax over the "logits"), yielding a distribution over words.

A consequence of the transition to address events is the ability to interpret language model output probabilities as estimates of world event probabilities (Li et al., 2022). We argue that this usage relies on two unjustified assumptions: it ignores the unique behavior of predictions, and it assumes equivalence between what is said and what is true. **Our position is that, without dedicated training of a separate probability estimator, the theoretically sound way to obtain world probabilities from a language model is to have the model explicitly report them in its output**—a method we describe as **second-order prediction**. Our position stands in contrast to common approaches that rely on the softmax-based generation probability of the output.

We demonstrate our claim with an example: consider a simple scenario with a biased coin with $p(H) > 0.5$ (see Fig. 1). Simple completion of a sentence—as in traditional language modeling— indicating the outcome, may reflect biases in the training data.[1] However, in many contexts, a user seeks the true $p(H)$: the probabilities of the underlying world events. For traditional language modeling to coincide with the event distribution, the training data must consist solely of faithful reports of events according to their world

[1]Department of Computer Science, Hebrew University of Jerusalem. Correspondence to: Eitan Wagner <eitan.wagner@mail.huji.ac.il>.

*Proceedings of the 43$^{rd}$ International Conference on Machine Learning*, Seoul, South Korea. PMLR 306, 2026. Copyright 2026 by the author(s).

---

[1]Gupta et al. (2025) show that models are generally biased towards `heads`, even when prompted that the coin is fair.

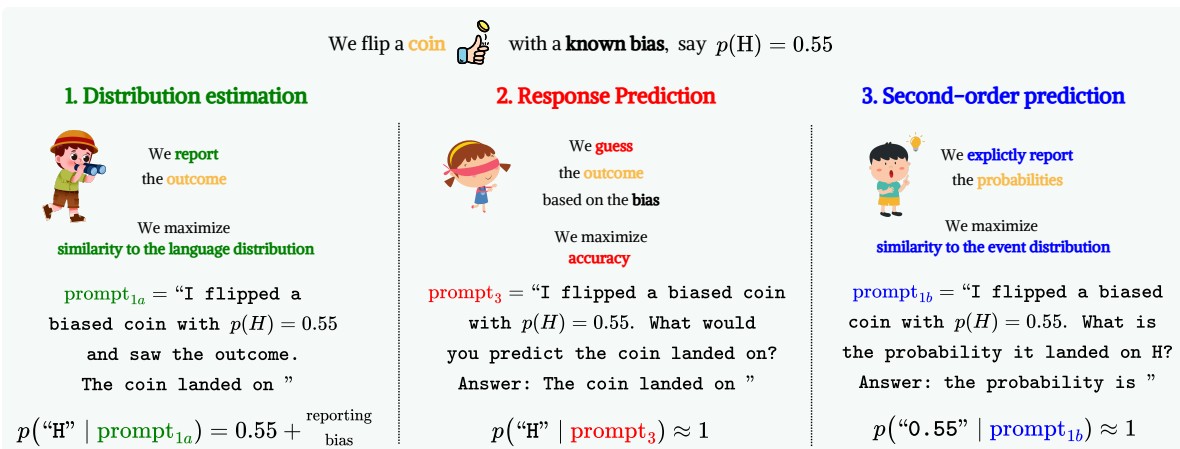

*Figure 1.* Example where distribution estimation (for language), response prediction, and target distribution estimation (for events), lead to different probabilities over a common output space. Models commonly have a strong bias to generate Heads, and utterances generated from the output *logprobs*—which estimate the data distribution—will reflect this additional bias (Case 1). Utterances that rationally predict an event (Case 2) should always select the mode. The event probabilities can be obtained by explicitly reporting them (Case 3).

distribution, which is not a common scenario (e.g., due to reporting bias). Alternatively, a user may wish for an accurate prediction about an unobserved outcome. The output distribution will thus be inherently different, as accuracy is maximized when the entire mass is on the mode (i.e., $p(H) = 1$). Indeed, calibrating the model with the data-generating distribution will lower the accuracy of prediction. This output distribution is expected when the model's objective considers the entire distribution. Importantly, this distribution is also expected if the data itself consists of predictions, even if the model is trained for standard language modeling. An example is when the input is an instruction to make a prediction (which should maximize accuracy). This example shows how subtle differences in the setting lead to inherently different target output distributions.

The paper mirrors the conceptual transition from words to world and its implications: First, we formally define the distinctions between the tasks of source distribution estimation, target distribution estimation, and response prediction (§2). We then discuss different training and inference settings for LMs (§3) and how they fit different use cases, in terms of corresponding formal tasks (§4). We present a probabilistic analysis of how different settings and inference strategies affect the expected output distributions, matching the different tasks (§5). Establishing the distinction between the seemingly similar tasks, we show that many works implicitly assume equality between the distributions in the different use cases, thereby ignoring the pitfalls on the way to world probabilities (§6). We conclude by discussing research directions for improving world modeling (§8).

## 2 Definitions: Estimation and Prediction

We begin with some formal definitions for the various tasks we discuss (see §C for a detailed glossary).

### 2.1 Distribution Estimation

The statistical task of **distribution estimation** involves approximating an unknown data-generating (or "source") distribution based on observed samples (Wasserman, 2013). More formally, in the conditional case, let $p_S(y \mid x)$ be a conditional **source distribution** we wish to estimate, where $x \in \mathcal{X}$ denotes an input and $y \in \mathcal{Y}$ denotes an output. The task of distribution estimation is to learn, based on observed data $S = \{(x_n, y_n)\}_{n=1}^N$ sampled from $p_S$, a mapping: $p_\theta : \mathcal{X} \to \Delta^{|\mathcal{Y}|-1}$. The resulting distribution estimator $p_\theta(\cdot \mid x)$ is an approximation of $p_S$.

Distribution estimators are typically fit to minimize the cross-entropy (or, equivalently, the Kullback-Leibler (KL) divergence), from the empirical distribution of the data $-\sum_{(x_i, y_i) \in S} \log p_\theta(y_i \mid x_i)$. In the optimal case, we get a model for which $p_\theta(\cdot \mid x) = p_S(\cdot \mid x)$, which completely recovers the data-generating distribution. We describe this as an **ideal estimator** and denote it by $p_{\theta^*}$.

**Target distribution estimation.** In some cases, we intend to estimate a distribution $p_T$ which is not identical to the source distribution, $p_S$. We call this **target distribution estimation.** The process in which the data from $p_S$ is used for improving a task over $p_T$ is sometimes referred to as **transfer learning** (Weiss et al., 2016). A simple case is when $p_S$ generates biased or corrupted samples of $p_T$ (Crammer et al., 2005). Knowing the relationship between the distributions allows us to learn about $p_T$ with data from $p_S$.

### 2.2 Response Prediction

As opposed to distribution estimation, **response prediction** is a task of returning a single output $y \in \mathcal{Y}$ for a given input $x \in \mathcal{X}$. We thus define a **predictor** as a function

$f_\theta : \mathcal{X} \to \mathcal{Y}$, which we learn from the data $S$ (Shalev-Shwartz & Ben-David, 2014).[2] The goal of prediction models is typically to minimize the expected loss defined for the output predictions, e.g., the misclassification error.

Given a dataset $\{(x_m, y_m)\}_{m=1}^M \sim p_S$, the predicted set $\{(x_m, f_\theta(x_m))\}_{m=1}^M$ can reflect a different distribution. For example, if $f_\theta$ is a deterministic function (e.g., a Bayes optimal classifier), the conditional entropy will be 0, which might not be the case in the original data. If, instead, a model randomly samples based on $p_{\theta*}$, then the original and predicted sets will follow the same distribution, and the entropy will be similar.

## 2.3 Confidence and Calibration

Response prediction and distribution estimation are often connected. Predictors can consist of a decision function on top of a learned distribution (Bishop, 2006), where we define a **decision function** $f_{\text{dec}}$ as a function from a distribution to a target $f_{\text{dec}} : \Delta^{|\mathcal{Y}|-1} \to \mathcal{Y}$. The predictor is thus $f_\theta = f_{\text{dec}} \circ p_\theta$.

**Example:** Consider the ideal model $p_{\theta*}$ and the decision function $f_{\text{dec}} = \text{argmax}$. The resulting predictor $f_\theta$ is known as the *Bayes optimal classifier* and is proved to be the one that minimizes the misclassification error, with respect to the distribution $p_S$ (Devroye et al., 2013).

When available, the probability assigned to a predicted output is often of interest, for example, when considering risks (Van Calster et al., 2019) or for building trust (Zhang et al., 2020). We describe this probability as the output's **confidence**. An important property of a model's confidences is **calibration**, which generally means that when a model outputs an answer with confidence $p$ then the model should be correct with approximately probability $p$ (Guo et al., 2017).

Notably, if the main focus is probabilities, target distribution estimation can be described as a prediction task, with $\Delta^{|\mathcal{Y}|-1}$, possibly rounded for discretization, as the output space. This task can therefore be performed by applying a decision over possible distributions in $\Delta^{|\mathcal{Y}|-1}$. We describe this setting as **second-order prediction** since the prediction is at a higher level of abstraction; a closely related framing is *probabilistic supervised learning* (Gressmann et al., 2018).

## 3 Praxis: LM Training and Inference

In this section, we describe common training and inference strategies used in contemporary LLMs and analyze their probabilistic interpretations. We show that chosen strategies directly impact the plausible interpretation of model outputs.

---

[2]A special case of response prediction, in which $\mathcal{Y}$ is a finite set, is also known as **classification**.

## 3.1 Probabilistic language modeling

A language model (LM) is traditionally defined as a distribution over finite strings (Shannon, 1948; Bahl et al., 1983). Formally, let $\mathcal{V}$ be an LM's vocabulary, composed of (sub)words $w \in \mathcal{V}$. An LM is then a distribution $p(\mathbf{w})$, where $\mathbf{w} \in \mathcal{V}^*$ is a sequence of words. Typically, language modeling is achieved through a next-word prediction model, which allows us to write the distribution autoregressively as $p(\mathbf{w}) = \prod_{t=1}^{|\mathbf{w}|} p(w_t \mid \mathbf{w}_{<t})$. We denote an ideal LM, i.e., one that perfectly models the data distribution, with $p_{\text{LM}}$.

Next-word prediction and language modeling are near equivalent, with an exception if next-word prediction assigns non-zero probability to an infinite sequence (disallowed by the traditional definition). Du et al. (2023) provide theoretical criteria, met by common LMs, for avoiding this case.

An LM also defines the conditional distribution $p(\mathbf{w}^{\mathbf{y}} \mid \mathbf{w}^{\mathbf{x}}) = \prod_{t=1}^{|\mathbf{w}^{\mathbf{y}}|} p(w_t^{\mathbf{y}} \mid \mathbf{w}^{\mathbf{x}} \circ \mathbf{w}_{<t}^{\mathbf{y}})$, where $\mathbf{w}^{\mathbf{x}}, \mathbf{w}^{\mathbf{y}} \in \mathcal{V}^*$ are word sequences, respectively representing input and output strings, $\circ$ is the concatenation function, and $p(\mathbf{w}^{\mathbf{x}}) > 0$.

## 3.2 Training Stages

Most modern LLMs go through a multi-stage training procedure (Touvron et al., 2023; Groeneveld et al., 2024).

**T1. Pretraining.** The first stage of fitting an LM is typically termed pre-training, and is a case of distribution estimation. Let $p_{\text{LM}}$ be an LM we wish to approximate, and $\{\mathbf{w}^n\}_{n=1}^N$ be a large set of strings sampled from it. We can consider each of these strings' time steps to define an input-output pair: $(w_t^n, \mathbf{w}_{<t}^n)$. We train a distribution estimator $p_\theta$ to minimize the cross-entropy on this data (§2.1). The resulting $p_\theta$ can be used as an LM (§3.1).

**T2. Adaptation/Fine-tuning:**

(a) **Supervised Fine-Tuning (SFT):** Modern models are explicitly finetuned to follow instructions (Zhang et al., 2024c; Touvron et al., 2023; Groeneveld et al., 2024).

Training datasets comprise instruction-following examples (Chung et al., 2022; Wang et al., 2023b;a). We term the set of possible instructions as the input space $\mathcal{X}$ and the set of possible responses as the output space $\mathcal{Y}$. Notably, the distribution of responses $p_{\text{res}}$ is not necessarily the same as the empirical distribution of the pretraining data, since it was constructed to include only helpful responses. As in pretraining, supervised instruction-tuning is done with a cross-entropy loss. The goal is to optimize $\theta$ to minimize the average cross-entropy over full responses $-\frac{1}{j} \sum_{i=1}^j \log p_\theta(y_i \mid x \circ y_{:i-1})$, for query+response pairs sampled from $p_{\text{res}}$.

(b) **Reward-based tuning.** A common additional step is training the model to align with human preferences. This

is typically done with a reward model trained using human feedback on outputs from the model (Ziegler et al., 2020; Chaudhari et al., 2024). The model is then trained to align with the reward model using reinforcement learning (RL) techniques (Rafailov et al., 2024; Shao et al., 2024). More recently, RL techniques for enhancing LLM reasoning capabilities through chains-of-thought and tool use are gaining popularity (OpenAI et al., 2024; Zhang et al., 2025b; Chen et al., 2025), leading to a new generation of models.

These methods optimize $\theta$ to maximize the **reward function**, $r : \mathcal{Y} \times \mathcal{X} \to \mathbb{R}$, representing the return of an output $y$ for an input $x$. Importantly, rewards are defined for output instances and not for a distribution. While the reward is maximized by placing all the mass on the highest rewarding response, in practice, a term is added to the reward function to penalize distance from the pretraining distribution.

### 3.3 Inference

For a given input $x$, we consider a set of outputs $\mathcal{Y}$. For example, if $x$ describes the event coin toss, we may consider the outputs $\mathcal{Y} = \{\texttt{heads}, \texttt{tails}, \times\}$, where $\times$ denotes an invalid response. As a user may wish to obtain probabilities for the possible outputs in $\mathcal{Y}$, we consider two general sources of probability: softmax-based and explicit report.

**I1. Softmax probabilities ("logprobs"):** Neural models typically contain a final layer, consisting of unnormalized log probabilities ("logits"). Applying the softmax to this layer yields next-token probabilities, typically given as "logprobs" (Vaswani et al., 2017). Notably, although the softmax probabilities are unavailable for closed models, they can often be approximated by sampling from the model (Kadavath et al., 2022). In this method, the probability for an output $y$ is proportional to the likelihood of generating its textual description $\mathbf{w}^{\mathbf{y}}$, given the input description $\mathbf{w}^{\mathbf{x}}$:

$$p(y \mid x) \propto p_{\text{LM}}(\mathbf{w}^{\mathbf{y}} \mid \mathbf{w}^{\mathbf{x}}) \tag{1}$$

The input $\mathbf{w}^{\mathbf{x}}$ is formatted in one of the following ways:

(a) **Naïve description:** The text input $\mathbf{w}^{\mathbf{x}}$ (in eq. (1)) can simply describe $x$. We denote the resulting probability $p_{\text{ND}}$.

> For example, if $x$ refers to Paris being the capital of a country, $x$ can be the sentence "`Paris is the capital of`". We can then look at the likelihood of "`France`" compared to the alternatives.

(b) **Zero-shot instruction:** Radford et al. (2019) show that pretraining alone gives models some capacity for many tasks. This capability is termed *zero-shot transfer*. By inputting an appropriate instruction prompt, the completion can be treated as a response. Formally, an instruction $\text{I}(x)$ is a transformation that formats the input $x$ as an instruction. We then use $\mathbf{w}^{\mathbf{x}} = \text{I}(x)$ in eq. (1). We denote the resulting probability by $p_{\text{ZS}}$.

> In the previous example, $\text{I}(x)$ can be something like: "`Q: What country is Paris the capital of? A: `".

(c) **Few-shot instruction:** Brown et al. (2020) further show that generative LMs are highly capable of following instructions based on a few examples. This method is known as *few-shot (in-context) learning*, as it only adds context with no explicit training. In this case, the query is preceded by a few examples of queries with expected responses. Similar to zero-shot instructions, we define a few-shot instruction $\text{FS}(x)$ as a transformation that formats the input $x$ as an instruction with a few examples. Then $\mathbf{w}^{\mathbf{x}} = \text{FS}(x)$ can be used in eq. (1). We denote the resulting distribution by $p_{\text{FS}}$.

> In the example, $\text{FS}(\mathbf{w}^{\mathbf{x}})$ can be something like:
> "`Q: What country is Madrid the`
> `capital of? A: Spain.`
> `Q: What country is Paris the capital`
> `of? A: `".

**Semantic probability.** The input and output strings $\mathbf{w}^{\mathbf{x}}, \mathbf{w}^{\mathbf{y}}$ have **referent events** $x \in \mathcal{X}$ and $y \in \mathcal{Y}$, respectively. A general challenge is that strings do not necessarily map naturally to events. For example, in addition to "`heads`", a model may give some mass to "`the coin landed on heads`", a phenomenon known as *surface-form competition* (Holtzman et al., 2021).

A common approach attempts to address *semantic probability* (Farquhar et al., 2024; Zhou et al., 2025). We can learn a mapping $\phi : \mathcal{V}^* \to \mathcal{Y}$ from string outputs to their referent event, such that a single referent event may be mapped to by many different strings, i.e., $\mathbf{w}^{\mathbf{y}_1} \neq \mathbf{w}^{\mathbf{y}_2}$ can map to $y_1 = y_2 \in \mathcal{Y}$. Then, instead of a single output likelihood $p_{\text{LM}}(\mathbf{w}^{\mathbf{y}} \mid \mathbf{w}^{\mathbf{x}})$, we look at a sum $\sum_{\mathbf{w}^{\mathbf{y}'} \in \mathcal{V}^*} p_{\text{LM}}(\mathbf{w}^{\mathbf{y}'} \mid \mathbf{w}^{\mathbf{x}}) \mathbb{1}_{\{\phi(\mathbf{w}^{\mathbf{y}'})=y\}}$ (Giulianelli et al., 2023; Meister et al., 2024).

While this approach can deal with simple surface-form competitions, it is not straightforward to generalize. For example, what should a string like "`probably heads`" match to? How should its similarity to "`heads`" be measured? Moreover, in more realistic cases, an input and output may contain many events, making the mapping nontrivial.

**I2. Second-order prediction:** Rather than using the output probabilities of the model to obtain response probabilities, we can instruct the model, possibly with few-shot examples, to explicitly report probabilities of different options. Importantly, this method directly addresses events and not their descriptions. Notably, unlike softmax probabilities, this method does not necessarily impose a specific format for the output. The output can be either numeric (where adherence to probability requirement must be externally verified) or verbal (e.g. "likely a but possibly b").

# 4  Use Cases: From Word to World

For many applications, models are used for textual tasks referring to reality (i.e., the "world"). Here we describe three common use cases, their connection to world probabilities, and their correspondence to training and inference strategies. Notably, while training involves *loss functions*, which are tailored for optimization, use cases involve *evaluation metrics*, which measure performance in downstream tasks (Raschka, 2024). Importantly, aligning losses and metrics may be challenging (Casas et al., 2018; Terven et al., 2025).

## 4.1  Word Completion

A common use of LMs is to randomly generate text continuing a prompt $\mathbf{w}^\mathbf{x}$ (Holtzman et al., 2020). Assuming we want the text to be similar to human-written text, a continuation $\mathbf{w}^\mathbf{y}$ should appear according to the ideal language modeling distribution $p_{\mathrm{LM}}$. An example of this is story generation (Teleki et al., 2025). In this case, there is no world of interest besides the words, which are the source distribution to estimate (2.1). The intended output distribution is naïve completion $p_{\mathrm{ND}}$ (Item (a)) as there is no intermediate instruction, and only pretraining (§3.2) is required.

## 4.2  World-related Responses

Perhaps the most common use of modern LMs is for answering world-related queries, e.g., in factual question-answering (Rajpurkar et al., 2016; Wei et al., 2024). The main concern is the quality of the response, in terms of truthfulness and avoiding harm. Scores are assigned accordingly. The rational choice, for optimal response quality, is to always generate the response with the highest score.[3] This is a case of response prediction (2.2). Since this case involves formatting the question as an instruction, inference should use the zero-shot distribution $p_{\mathrm{ZS}}$ (Item (b)) or, preferably, the few-shot distribution $p_{\mathrm{FS}}$ (Item (c)). Training for instructions (Items (a) and (b)) will be beneficial.

## 4.3  World Modeling

In many cases, a user seeks the probability of world events. The user might want a confidence score for a single prediction, e.g., for risk assessment (Van Calster et al., 2019), or to infer a full unbiased distribution of possible events (Nafar et al., 2025). In this case, we look at the event distribution $p_{\mathrm{E}}(y \mid x)$, for $x \in \mathcal{X}$ corresponding to text input $\mathbf{w}^\mathbf{x}$ and $y \in \mathcal{Y}$ corresponding to a possible text output $\mathbf{w}^\mathbf{y}$. This is a case of target distribution estimation (2.1).

---

[3]In reality, humans tend to add randomness to their choices, even in prediction cases. See *prospect theory* (Vulkan, 2000; Shanks et al., 2002). Some cognitive frameworks, such as RSA (Goodman & Frank, 2016), model agents with different levels of "rationality" according to how concentrated the distribution is.

**Output probabilities.** When using output probabilities from a language model, the difference between the word and world distributions must be accounted for. Instruction-based inference (i.e., $p_{\mathrm{ZS}}$ or $p_{\mathrm{FS}}$) with a question about events gives a distribution reflecting choices of optimal responses, which is generally not the true world-event distribution. Using $p_{\mathrm{ND}}$ without an instruction should match the language modeling distribution, and will reflect the world-event distribution only if the language modeling distribution is unbiased. Setting aside some toy settings, this is not the case in general. For example, Paik et al. (2021) show that LMs are biased towards describing bananas as green, presumably since yellow bananas are commonly described without their color. Additionally, this type of inference requires a mapping between words to world events, which is non-trivial, especially for large output spaces (§3.3).

**Second-order prediction.** Asking for explicit probabilities, as in second-order prediction (3.3), can, if successful, obtain unbiased probabilities. The user can decide on the granularity of the probabilities, from numerical outputs to natural language descriptions (e.g., "probably a but maybe b"). This method is highly dependent on the quality of the model, as it requires inference about a distribution other than the training data. Therefore, in practice, extensive adaptation training is used (Items (a) and (b)).

# 5  Analysis: Prompts and Ideal Distributions

In this section, we propose a formal Bayesian framework for the generation process of textual responses, which takes into account different goals. We start with definitions and move to address three cases that correspond to different data-generating distributions, represented by three ideal conditional LMs. Our analysis shows how different input prompts can lead to different ideal output distributions.

## 5.1  Definitions

We use the term **ground** for the set of propositions that are common knowledge (Green, 2021). This includes both explicitly stated propositions and propositions stipulated based on common knowledge or some inference process. The text is assumed to be generated by an agent, which we describe on the basis of the belief-desire-intention (BDI) framework (Bratman, 1987; Zuppiroli et al., 2025). The agent observes the **world**; forms **beliefs** which are the agent's knowledge; has **desires** which are its general goals; and finally derives **intentions** which are specific plans for actions.

We use the following (simplified) setting: an input string $\mathbf{w}^\mathbf{x}$ sets the ground. Possible output strings $\mathbf{w}^\mathbf{y}$ can be generated by some agent, whose beliefs and desires can be established from the ground. We define the agent's beliefs $b^\mathbf{y}$ as a set of propositions the agent holds—these may be events (e.g., "$y$

| Use case | Task type | Training strategy | inference strategy | Data |
|---|---|---|---|---|
| **Completion distribution** | distribution estimation | pretraining | logit probabilities + naive completion | language utterances |
| **Response distribution** | prediction | instruction tuning | logit probabilities + zero-shot instructions | generated by predictors |
| | | | logit probabilities + few-shot instructions | |
| **Event distribution** | target distribution estimation | pretraining | logit probabilities + naive completion | fully observed and faithful reports |
| | | instruction tuning | second-order prediction | data for transfer learning |

*Figure 2.* Probability interpretations and their mapping to task type (§2), training and inference (§§ 3.2 and 3.3), and relevant data (§5).

occurred") or probabilistic statements (e.g., "$p(y \mid x) = q$"). Its desire $d^{\mathsf{y}}$ is a random variable $f$ drawn from a space of (possibly stochastic) functions from beliefs to strings $D^{\mathsf{y}} = \{f_n\}$, $f_n : b^{\mathsf{y}} \rightarrow \mathcal{V}^*$. $p(f \mid \mathbf{w}^{\mathsf{x}})$ describes the probability of applying the function $f$ given the ground $\mathbf{w}^{\mathsf{x}}$. We further assume here that $\mathbf{w}^{\mathsf{x}}$ includes a reference to some world event, denoted by $x$, in addition to other possible information, and that $\mathbf{w}^{\mathsf{y}}$ has a unique referent event $y$. The agent intends (and executes) an action about $y$ based on its beliefs $b^{\mathsf{y}}$, which include propositions the agent holds about the world, and desires $D^{\mathsf{y}}$, which can include predicting unobserved outcomes or faithfully reporting observed ones.

In the coin toss example, $\mathbf{w}^{\mathsf{x}}$ states a coin toss event $x$ and possibly adds information, such as whether the outcome $y \in \{\texttt{heads},\texttt{tails}\}$ was observed. The agent states whether the outcome was $\texttt{heads}$ or $\texttt{tails}$.

We show that for the action distribution to match the world distribution, bias between the world and beliefs about it, and between beliefs and intentions, should be minimal.

### 5.2 Case 1: Multiply Describable Outcomes

In the simple case, we consider a set of possible output strings $\mathbf{w}_1^{\mathsf{y}}, \ldots \mathbf{w}_k^{\mathsf{y}}$ referring to a single event $y$, with a non-instruction input prompt $\mathbf{w}^{\mathsf{x}}$. For example, the possible output strings can be "$\texttt{The coin landed on Heads}$" and simply "$\texttt{Heads}$".

Here, the output probabilities reflect manners of expression. If the ground $\mathbf{w}^{\mathsf{x}}$ imposes no restrictions, the LM should reflect the frequency of use. This aligns with the original language modeling definition, used for modeling statistical patterns in language usage (Shannon, 1951). The similarity between the output probabilities $p_{\mathrm{LM}}(\cdot \mid \mathbf{w}^{\mathsf{x}})$ and the event probabilities $p_{\mathrm{E}}(\cdot \mid \mathbf{w}^{\mathsf{x}})$ depends on the degree of bias between the observed events (which are linguistic here) and the belief (which reflects world events).

### 5.3 Case 2: Observed Outcome

A different situation is when we consider possible responses that refer to different outcomes. For simplicity, we assume that $\mathbf{w}^{\mathsf{y}}$ is the only way to express $y$. We also assume that $\mathbf{w}^{\mathsf{x}}$ contains, possibly implicitly, an instruction $\mathrm{I}(x)$ for the agent. How does the probability $p_{\mathrm{LM}}(\mathbf{w}^{\mathsf{y}} \mid \mathbf{w}^{\mathsf{x}})$ relate to $p_{\mathrm{E}}(y \mid x)$? The relationship depends on the difference between the agent's belief and its intention.

Assume the agent observes the outcome, i.e., $\mathbf{w}^{\mathsf{x}}$ states that $y \in b^{\mathsf{y}}$, and it always faithfully reports that outcome, i.e., $d^{\mathsf{y}} = \{f(y) = \mathbf{w}^{\mathsf{y}}\}$. Here, the average belief aligns with the intention, and we get $p_{\mathrm{LM}}(\mathbf{w}^{\mathsf{y}} \mid \mathbf{w}^{\mathsf{x}}) = p_{\mathrm{E}}(y \mid x)$. In practical scenarios, an agent does not always faithfully report $y$, and the resulting distribution may be biased. This occurs when the agent assumes $y$ to be obvious (e.g., the banana is yellow), when the agent is deceptive, or when the agent has noisy observations. Therefore, we cannot assume the output $p_{\mathrm{LM}}$ reflects an unbiased world distribution.

In the coin example, $\mathbf{w}^{\mathsf{x}}$ states an agent saw the outcome of a coin toss. If it also states that the agent always reports the outcome, the distribution should reflect the coin bias (=event distribution). Otherwise, reporting bias may be present.

### 5.4 Case 3: Unobserved Outcome

A different case is when the agent has no knowledge of whether the event $y$ happened or not (according to the ground $\mathbf{w}^{\mathsf{x}}$). Here too, we assume $\mathbf{w}^{\mathsf{y}}$ is the only way to express $y$. In this case, besides describing the input event, $\mathbf{w}^{\mathsf{x}}$ can include an instruction to the agent, implying that it must make a prediction. The predictors that the agent can apply are those in $D^{\mathsf{y}}$. Thus, we write:

$$p(\mathbf{w}^{\mathsf{y}} \mid \mathbf{w}^{\mathsf{x}}) = \sum_{f \in D^{\mathsf{y}}} p(f \mid \mathbf{w}^{\mathsf{x}}) \cdot p(f(b^{\mathsf{y}}) = y) \quad (2)$$

Assume the agent's belief includes some estimate of the event probability, which it can in principle report (i.e., $\mathbf{w}^{\mathbf{x}}$ implies $\hat{p}(y \mid x) \in b^{\mathbf{y}}$, where $\hat{p}$ is the agent's estimate). For $p(\mathbf{w}^{\mathbf{y}} \mid \mathbf{w}^{\mathbf{x}})$ to match $p(y \mid x)$, two conditions must hold jointly: (i) **correct belief**—the agent's estimate matches the event probability, $\hat{p}(y \mid x) = p(y \mid x)$; and (ii) **faithful reporting**—the desire distribution $p(f \mid \mathbf{w}^{\mathbf{x}})$ places its mass on predictors $f$ that sample $y$ with $p(f(b^{\mathbf{y}}) = y) = \hat{p}(y \mid x)$.[4] In practice, both conditions can fail. Beliefs can diverge from event probabilities, sometimes systematically— a finite agent has access only to a subset of evidence, and the available evidence may itself be skewed. Faithful reporting is also not rational under standard prediction objectives, since an agent optimizing accuracy should put probability 1 on the most likely outcome. A significant mismatch is therefore likely on either front.

Recall the coin example and assume $\mathbf{w}^{\mathbf{x}}$ states that a coin was tossed and that an agent who did not see the outcome must predict it. The probability for prediction `heads` will be affected by the predictor's goal (e.g., if the objective is accuracy, then it can almost always choose the mode).

In §A we further discuss how the different training and inference approaches (§3) affect the agent's choice of $p(f \mid \mathbf{w}^{\mathbf{x}})$ (=its desire). The resulting action can diverge from the world distribution, even with unbiased beliefs.

## 5.5 Second-Order Prediction

If the agent's belief includes the event probability (i.e., $\mathbf{w}^{\mathbf{x}}$ implies $p(y \mid x) \in b^{\mathbf{y}}$), and the ground $\mathbf{w}^{\mathbf{x}}$ asks it to report this probability rather than predict an instance, we have second-order prediction. Crucially, the bias identified in Case 3 disappears: even an agent optimizing for accuracy can faithfully report the believed probability, because reporting the probability is the accurate action. The output distribution matches the world distribution as long as the agent's probabilistic belief is unbiased.

## 6 Pitfalls in Computing World Probabilities

Suppose we want world probabilities and all we have is textual data. We view the inference process in three levels— words, responses, world—and argue that the apparent similarity between them has led many works to implicitly equate distributions that are in fact distinct. In what follows, we show that clarifying the differences can explain hitherto unexplained results in the literature in some cases. In other cases, we find that the difference between these definitions was seemingly overlooked, leading to unwarranted claims or expectations. We note that these works also have many mer-

---

[4]A correct belief with faithful reporting is sufficient for $p(\mathbf{w}^{\mathbf{y}} \mid \mathbf{w}^{\mathbf{x}}) = p(y \mid x)$; other options might yield the same marginal by coincidence, but there is no reason to expect this in general.

its and contain meaningful experiments, which we do not discuss. Our goal is to demonstrate how the formalization of the different interpretations of the LM output distributions can inform ongoing discussions, uncover misconceptions, and suggest more theoretically sound ways of addressing language model probabilities.

### 6.1 Estimation $\neq$ Prediction

One hurdle we must overcome is going from the word distribution to a model's output distribution, and, similarly, from a model's output distribution to the world distribution. Both steps compare distribution estimation and the distribution of responses.

**Completion $\neq$ Response.** Consider the following prompts, and consider the probability of the next word being `is` or `are` (as in Hu & Levy, 2023):

(a) "The keys to the cabinet"
(b) "Here is a sentence: The keys to the cabinet... What word is most likely to follow?"

Despite the apparent similarity, the probabilities are essentially different: $p(\text{are} \mid \text{prompt (a)})$ comes from the completion distribution which is a distribution estimation task, whereas $p(\text{are} \mid \text{prompt (b)})$ comes from the response distribution, which is a prediction task. While $p(\text{are} \mid \text{prompt (a)})$ should be similar to the frequency in the training data, $p(\text{are} \mid \text{prompt (b)})$ should be close to 1 if the completion "are" is the most frequent in the data.

Despite being unjustified, this assumption—that prompt (a) and prompt (b) yield the same probabilities—is not rare. For example, Hu & Levy (2023) compare the quality of predictions in these two cases, explicitly stating that "a model that perfectly performs the metalinguistic task posed in prompt (b) should have identical probability distributions over the next word given prompts (a) and (b)." Their findings about a performance gap should therefore not be interpreted as a deficiency of the LM, but rather as an inherent discrepancy between the two distributions.

Similarly, Wagner et al. (2024b) design tests for consistency of span distributions by comparing equivalent factorizations of the joint probability. The factorization involves masked language modeling (MLM), which, for autoregressive models, is estimated using a prompt instructing to fill in a missing word (Sections 4 and 5.1). As MLM is a distribution estimation task (like prompt (a)) and instruction-following is a prediction task (like prompt (b)), equating the terms is unwarranted. Here too, the results showing different trends are expected.

Additional cases of this confusion are when assuming highest-likelihood text generations are of the highest quality

(see §B.1) and when treating Shannon's guessing game as equivalent to autoregressive language modeling (see §B.2).

**Response ≠ Event.** Consider the following question with two possible answers (Yona et al., 2024):

> (Q) "When was Mark Bils born?"
> (A1) "March 22, 1958."
> (A2) "I think he was born on March 22, 1958, but not sure."

The answer (A1) is worded more conclusively than the answer (A2). Assume the model $M_1$ generates (A1) and the model $M_2$ generates (A2), given the question (Q). Should we expect $p_{M_1}(\text{A1} \mid \text{Q})$, when the other option is an answer with a different birth date, to be significantly higher than $p_{M_2}(\text{A1} \mid \text{Q})$, reflecting the difference in confidence? not necessarily. If the objective is accuracy, then both models should **always** generate the same birth date as long as there is no better option. The confidence reflects the probability of being correct, which is a reported event distribution, whereas the generation probability reflects what the model should output, which is a response distribution.

This distinction has often been overlooked. For example, Yona et al. (2024) compare the decisiveness in a model's response to its intrinsic confidence, defined as the generation probability. The paper explicitly describes a large gap as "unnecessary hedging" (Section 2), and concludes that identified gaps show that LLMs are poor at "faithfully conveying uncertainty" (Conclusion). As we argue that generation probabilities should not be interpreted as confidence scores, the resulting gap is unsurprising, and the conclusion unwarranted. Similarly, Liu et al. (2023) investigate the gap between output probabilities with a query and internal probing probabilities. Here too, response probabilities are assumed to ideally match correctness probabilities. More generally, this confusion commonly arises when measuring calibration of instruction-tuned LMs through their softmax probabilities (see §B.3 for examples).

The common finding in these papers—*overconfidence*, where output probabilities exceed correctness probabilities—serves as empirical evidence for our framework. Under a response-prediction objective, the output distribution should concentrate on the model's best guess, which means generation probabilities are systematically higher than the correctness probability of that guess. What the literature often labels a calibration failure is, on our view, a category error: the two distributions were never supposed to coincide.

### 6.2 Population ≠ Event

The previous subsections compared distributions at the level of an individual agent: word completions vs. responses, and responses vs. correctness. Here we move to the population

level, where the relevant comparison is between the distribution of reports across text-generating agents and the distribution of events in the world.

Consider the following question and two candidate response distributions:[5]

> (Q) "Is Earth flat or round?"
> (D1) 80%: "Earth is round"; 10%: "Earth is flat"; 10%: "Not sure";
> (D2) 100%: "Most of the population believes Earth is round. Despite scientific evidence for this belief, a non-negligible part of the population believes Earth is flat."

Response (D2) avoids two challenges arising in (D1):

**Challenge 1: Sampling ≠ plurality.** In any given interaction, a model sampling according to (D1) will voice one position with full confidence and leave the others out. In roughly 10% of interactions, the model will confidently assert that Earth is flat, possibly addressing other opinions as brainwashed. The aggregated response (D2) preserves the same information without this side effect. Importantly, this argument does not depend on the population being wrong: even if every opinion is treated as legitimate, sampling hides the plurality it is supposed to represent. Both issues mirror failure modes at the individual level (§5.4): biased reports correspond to faithful reporting failing, misinformed beliefs to incorrect belief.

This point has often been ignored. Sorensen et al. (2024) suggest three types of pluralism in LM generations, including *distributional pluralism*, in which generated answers reflect the distribution of positions in the population.[6] Works that adopt this approach as a default sample from (D1) by construction, losing the pluralism at the sample level. Lake et al. (2024) document an alternative (also named by Sorensen et al. (2024)): RLHF models shift toward *Overton pluralism*, aggregating the relevant positions into a single response—closer to (D2), which we view as the more rational mechanism.

**Challenge 2: Population belief ≠ event distribution.** (D1) may not reflect an informed estimate of $p(y \mid x)$. Raw textual data is generally a biased sample of population beliefs: reporting bias (Gordon & Van Durme, 2013), the tendency to describe the unusual (Paik et al., 2021), and over-representation of dominant perspectives (Santurkar

---

[5] Percentages are based on a 2022 survey of American adults (Hamilton, 2022).

[6] They note that distributional pluralism, as commonly implemented, assigns **higher** probabilities to more frequent positions—a property they identify as a limitation. This concern arguably prioritizes diversity over accuracy.

et al., 2023) all skew which positions get expressed and how often. A well-designed survey controls for this through sampling and weighting, but a distribution induced by naïve text data does not. Furthermore, the population itself may be misinformed and should not be mimicked. A model with access to scientific evidence should not assign 10% probability to a flat Earth just because 10% of people believe so. Rather, the model should express informed uncertainty—i.e., an estimate of the world event probabilities.

Some existing work treats the distribution of reports as itself the correctness distribution, particularly in annotator-disagreement settings, where a population of annotators is construed as predictors of an underlying label. Baan et al. (2022) posit that a model assigning probabilities according to the proportion of annotator labels is calibrated by definition—conflating the proportion of annotators choosing $y$ with the probability that $y$ is correct. Similarly Meister et al. (2025) construct benchmarks aligning LLM output distributions to human variation under essentially the original assumption. See §B.4 for further discussion on the relationship between the population and correctness.

## 7  Alternative Views

The alternative view treats the softmax logprobs as reliable probability estimates for world events. This view is divided into two positions: one is that distribution estimation on texts, possibly with simple transformations, can be reliably used for world events (as in §4.3). The other is that the probabilities can be used even when instructed to predict an answer (§4.2). As we show in §6, these views are common even in respected venues, even if unjustifiably.

## 8  Discussion and Conclusion

**Diversity and mode collapse.** LLMs are often construed as agents in a Markov Decision Process (MDP), with textual states and actions (Ziegler et al., 2020). Li et al. (2024) demonstrate how RLHF shifts LLMs from world models to agents, arguing that it promotes planning (especially with long outputs), at the expense of modeling. More generally, SFT and RLHF have been observed to reduce diversity, a phenomenon termed "mode collapse" (O'Mahony et al., 2024; Jiang et al., 2025; GX-Chen et al., 2026). Our analysis shows that this may be due to implicitly or explicitly changing the LLM's goal, which was shown to affect output probabilities (cf. Kalai et al., 2025). Formatting prompts as questions directs LLMs into prediction tasks, where mode collapse may be optimal.

**Practical takeaways.** Among the three common uses of LLMs outlined in this paper, second-order prediction is the only one aligned in its goals with obtaining event probabilities. Notably, unlike common RL methods (see Item (b)), second-order prediction rewards a full distribution, without altering the general framework. It is also naturally suited to cases where the desired output is a stochastic policy (e.g., "choose a random number"), since the policy can be reported directly rather than relying on outputs to be calibrated. We do not claim current models necessarily perform second-order prediction well. Verbalized probabilities are often poorly calibrated (Xiong et al., 2024)—though better than logprob-based estimates (Tian et al., 2023). Expressiveness is also a challenge: verbally reporting a distribution limits the granularity and structure that can be conveyed, and for long outputs (e.g., story generation), listing options with scores (as in Zhang et al. (2025a)) quickly becomes prohibitively long.

Crucially, these are limitations of the model's reasoning or knowledge, not of the objective. Improving second-order prediction means improving the model; by contrast, removing biases in logprob-based methods (e.g., removing a tendency toward `heads`) would require changing the data distribution or the loss itself. There is no analogous fundamental obstacle for second-order prediction and improving such reports falls squarely within the scope of standard post-training (Items (a) and (b)).

**Reasoning as a path forward.** A shared route to addressing limitations of second-order prediction is to leverage the reasoning capabilities of modern models. Rather than asking a model to verbalize a distribution directly, reasoning can be used to fit explicit formal models—Logistic Regression (Wagner et al., 2024a), Bayesian networks (Nafar et al., 2025)—whose parameters or structure the model infers from the query. The same approach scales to large output spaces: a hierarchical reasoning process can first identify a high-level attribute (e.g., persona or topic) and then conditionally generate content (Wolfson et al., 2025), preserving diversity that a flat verbal distribution would lose. Broadly speaking, we argue that robust world modeling should be pursued by generating content that expresses calibrated probabilities, rather than by calibrating the token-level probabilities over alternative outputs.

**Closing remarks.** While LLMs are used in diverse ways, only a few works have attempted to formally define their goals, resulting in a sizable body of work that unwarrantedly assumes different settings should result in similar distributions. Our work sets firmer ground for this topic and identifies common pitfalls when applying output distributions for different purposes. We thus advise caution when combining results from classic and aligned models, as they were developed to reflect fundamentally different distributions.

## Acknowledgments

This research was supported by grants from the Israeli Ministry of Science and Technology, the Council for Higher Education, and the Israel Science Foundation (Grant No. 2424/21). The authors are grateful to Clara Meister and Tiago Pimentel for their significant and highly valuable contributions to this work. The authors would also like to thank the NLP reading group at the Hebrew University for valuable feedback and discussions.

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

# A    Controlling the Desire

How can we affect the distribution in which the agent selects the predictor $p(f \mid \mathbf{w}^{\mathbf{x}})$ (=its desire)? This can be done either by training the agent or by setting the ground.

Writing $p(f \mid \mathbf{w}^{\mathbf{x}}) \propto p(f) \cdot p(\mathbf{w}^{\mathbf{x}} \mid f)$ shows three possible ways to affect the choice of the predictor: (1) Training the prior probability $p(f)$. This is the approach in agent-based training, such as RLHF (see Item (b)). ; (2) Training the conditional probability $p(f_\theta \mid \mathbf{w}^{\mathbf{x}})$, for an appropriate input string. This is the approach in supervised fine-tuning for instructions (see Item (a)); and (3) Changing the ground so as to affect the probability. This can be done by raising the likelihood of this prompt given the predictor $p(\mathbf{w}^{\mathbf{x}} \mid f)$. An example of this is few-shot in-context learning (Item (c)), where $\mathbf{w}^{\mathbf{x}}$ includes example responses attributed to the responding agent. Replacing $\mathrm{I}(x)$ with $\mathrm{FS}(x)$ will raise the probability of a predictor $f$ that follows the examples.

In conclusion, different input formats can affect the desire, leading to a possible discrepancy between the agent's beliefs and intentions. The resulting action will diverge from the world distribution, even with unbiased beliefs.

# B    Additional Examples

## B.1    Text Generation

In some examples, probabilities that should represent distributions are assumed to represent predictions. For example, Holtzman et al. (2020) discuss the curious phenomenon that text generated by the highest likelihood tends to have low quality. As Meister et al. (2022) note, selecting the highest likelihood yields predictable strings with substantially less information than the distribution over natural strings. Arguably, text generation by highest likelihood treats language generation as a response prediction task, ignoring the fact that text completion is a task that should use distribution estimation (see §4.1).

## B.2    Shannon's Guessing Game

In his famous work, Shannon (1948) estimated the entropy and redundancy of the English language, implicitly defining the language modeling distribution. Shannon (1951) further derived empirical bounds for the entropy of the English language through a guessing game. In one version of the game, a person is randomly given texts with a previous (limited) context and is instructed to guess what the next letter is. The guesser continues guessing until they guess the correct letter, and the number of attempts is recorded. Based on his results, Shannon derived upper and lower bounds on English.

Despite the similarity between the guessing game and language modeling, they are fundamentally different in their objectives: language modeling is an *estimation task*. The goal is to represent the *probabilities* of occurrence of sequences of words or characters in natural language. In Shannon's guessing game, on the other hand, the goal is to give an optimal *prediction* of the real character and not to estimate its probability. The guesser is assumed to always choose the highest-ranking character in the distribution. For this reason, the results from the game were used to derive bounds and not for direct estimation.

Works that address the game as an autoregressive language modeling (e.g., Zouhar et al., 2024) ignore the predictive nature of the game, thus confusing a completion distribution with a response distribution (see 6.1).

We also note that it is incorrect to assume that the ranking in a prediction task is equivalent to Shannon's game. While Shannon assumes the guesser to guess based on the rank in some implicit language modeling, the probabilities of a prediction model have no guarantees beyond the identity of the highest-ranking prediction.

## B.3    Calibration

Consider a question for which a model outputs an answer $a$ with confidence $p > 0.5$. Calibration implies that the probability of $a$ being correct equals $p$ (§2.3). However, this does not mean the LM should generate the answer $a$ with frequency $p$, since maximal accuracy is achieved by generating $a$ with probability $1$. However, many works address calibration of generative LMs by assessing the generation probabilities, assuming the probabilities should ideally reflect confidence. For example, Kalai & Vempala (2024) derive a lower bound on the hallucination rate for calibrated models, with the assumption that calibration is appropriate in the generation setting. In another example, Zhang et al. (2024a) explicitly assume that fitting answers that appear more frequently in the corpus should elicit calibration. Many other works (Kadavath et al., 2022; Jiang et al., 2021; Zhu et al., 2023; Dhuliawala et al., 2022; Yang et al., 2025) treat the, possibly normalized, output probabilities

as confidence scores. The general finding in these papers, that LMs are poorly calibrated and that instruction tuning hurts calibration, is therefore not surprising. Notably, Tian et al. (2023) compare output probabilities to verbally expressed ones and find that verbal probabilities are better calibrated. While this result aligns with our analysis, we argue that the distinction is not merely experimental but rather reflects distinct underlying distributions.

### B.4  Human Disagreement

We provided examples of cases in which the model has access to information that explicitly addresses and refutes a common belief (fact-checking). A similar argument can be made for the case of multiple judgments. Consider a case where multiple independent judges are prompted with an identical question, and some answer $a$ is generated with proportion $p > 0.5$. If mistakes behave like white noise, the correctness probability of the majority vote will increase with the number of judges, with a probability that goes to 1 (McLennan, 1998), thus in this case, $p$ underestimates the correct probability. This phenomenon is known as "wisdom of the crowd" and is relevant to some domains but not others (Kuncheva et al., 2003). [7]

The works critiqued in §6.2 ignore the domain dependence. Baan et al. (2022) treat annotator-proportion as calibration by definition. In later work, Baan et al. (2024) acknowledge the difference between confidence and human label variation and discuss the merits of both measures. Meister et al. (2025) similarly assume the alignment between LLM output distributions and human variation is itself desirable. Notably, these two latter papers discuss calibration and variation of the output probabilities, thus confusing them with correctness probabilities (§6.1) and the frequency in the data (§6.1).

## C  Glossary

**0-1 loss:** Also called *misclassification rate* (Shalev-Shwartz & Ben-David, 2014). This is the expected number (given a target distribution and a predictor) of incorrect predictions. The misclassification rate is closely connected to accuracy, which is the expected number of correct predictions.

**Belief:** An agent's knowledge. We define the agent's beliefs as the set of events $b^{\mathrm{y}} = \{y_n\}$ the agent knows.

**Calibration:** Calibration is a measure of the quality of the output confidence of a model. A model is said to be fully calibrated if, for any $0 \le p \le 1$, the expected proportion of correct predictions in the set of predictions with confidence $p$ is exactly $p$ (Guo et al., 2017). In practice, predictions are binned by confidence intervals, and approximate calibration scores are measured (Naeini et al., 2015).

**Conditional Distribution:** We denote the conditional distribution of $Y$ given $X$ by $p_{Y|X}$. Since we only deal with conditional distributions of this sort, we simply use $p$. It is common to use the notation $p(y \mid x) \coloneqq p_{Y|X}(x, y)$.

**Confidence:** In addition to a specific output $y \in \mathcal{Y}$, a *predictor* can output an estimated probability $p$ for $y$ being the correct output. $p$ is called the confidence score.

**Decision function:** We define a **decision function** $f_{\mathrm{dec}}$ as a function from a distribution to a target $f_{\mathrm{dec}} : \Delta^{|\mathcal{Y}|-1} \to \mathcal{Y}$.

**Desire:** An agent's general goals. In our setting, we formalize a desire $d^{\mathrm{y}}$ as a random variable $f$, drawn from a space of functions from beliefs to strings $D^{\mathrm{y}} = \{f_n\}$, $f_n : b^{\mathrm{y}} \to \mathcal{V}^*$.

**Distribution estimation:** The task of approximating a (conditional) *distribution* $p_S$. This is an application of *density estimation* to discrete random variables.

**Distributional divergence:** This includes all functions that measure distances or divergences between probability distributions. These include proper distance functions, such as Total-variation distance and JS-divergence, and also measures like JS- and KL-divergence and cross-entropy (Tsybakov, 2008). These measures are minimized when the two compared distributions are identical.

**Ground:** We use the term ground for the set of propositions that are considered true (Green, 2021). The ground can be set in an input prompt $\mathbf{w}^{\mathrm{x}}$.

---

[7]See Zhang et al. (2024b) for analysis of different types of disagreement between human labels and the effect of a majority vote.

**Ideal estimator:** A model that perfectly represents the *source distribution*. This means the *distributional divergence* between $p_\theta$ and $p_S$ is minimal. Ideal estimation is generally unattainable but can serve as a proxy for the intended behavior.

**Input variable:** We denote an input random variable by $X$, with possible values $x \in \mathcal{X}$.

**Intention:** Specific plans for actions.

**Language modeling:** An LM is traditionally defined as a distribution over finite strings (Shannon, 1948; Bahl et al., 1983). Formally, let $\mathcal{V}$ be an LM's vocabulary, composed of (sub)words $w \in \mathcal{V}$. An LM is then a distribution $p(\mathbf{w})$, where $\mathbf{w} \in \mathcal{V}^*$ is a sequence of words.

**Output variable:** We denote an output random variable with $Y$, with possible values $y \in \mathcal{Y}$.

**Predictor:** This is simply a function $f_\theta : \mathcal{X} \to \mathcal{Y}$. A predicted value $f_\theta(x)$ is called a *prediction*.

**Referent Event:** A referent event is an occurrence in the world to which an utterance refers. We denote the event referred to by an utterance $\mathbf{w}^{\mathbf{x}}$ by $x$.

**Response prediction:** This is the task of outputting a value $y \in \mathcal{Y}$, given an input $x \in \mathcal{X}$ (Shalev-Shwartz & Ben-David, 2014). When $\mathcal{Y}$ is a discrete, finite set, this task is often called *classification*.

**Reward function:** A reward function (or simply a *reward*) is a function $r : \mathcal{X} \times \mathcal{Y} \to \mathbb{R}$ representing the return of an output $y$ for an input $x$. Here too, we define $r_x(y) := r(x, y)$. In many cases, a modified reward function is used, in which a term is added to penalize a distribution that is far from the one resulting from pretraining.

**Source distribution:** the (conditional) distribution from which training data is generated. We denote this by $p_S$.

**Second-order prediction:** A prediction setting in which the output is not a single instance $y \in \mathcal{Y}$ but rather a distribution over the possible instances $p \in \Delta^{|\mathcal{Y}|-1}$.

**Target distribution estimation:** Target distribution estimation is the task of estimating a distribution $p_T$ which is not identical to the source distribution, $p_S$, from which the data is sampled.

**Transfer Learning:** A process in which data from $p_S$ is used for improving a task over $p_T$ (Weiss et al., 2016).

**World:** We use this term to describe events that occur and may or may not be known to an agent.

