# OpenReview forum: "Position: Express Your Doubts — Probabilistic World Modeling Should Not Be Based on Token *logprobs*"
_ICML.cc/2026/Position_Paper_Track — ICML 2026 Position Paper Track regular_

### Official Review · Reviewer_UPnH · 2026-03-04

**Significance:** 3
**Argument Clarity:** 3
**Rating:** 3
**Confidence:** 3

**Questions:**

- In section 5, a theoretical framework for the generation of texts is established. Why isn’t this framework also used to analyse common pitfalls in section 6? Do the authors see uses of the framework proposed in section 5 beyond the conducted analysis and the scope of this paper?

- In section 5 Case 2, the use of logprobs for world modeling under observed outcomes is discarded as fundamentally flawed since the agent might be deceptive or suffer from noisy observations. But couldn’t similar reasons also bias second-order-prediction? Why is second-order-prediction not flawed due to practical difficulties in the same way?

- In section 6, the authors claim that the results from Yona et al. (2024), who found a gap between the decisiveness of the responses of LMs and their internal confidences, are unsurprising and the conclusions unwarranted. I follow the argument that these results might be expected, but isn’t the conclusion that LMs are poor at conveying confidence still true? Aren’t Yona et al. (2024) in fact reaching a similar conclusion as the presented paper in this point, only coming from an empirical angle?

**Alternative Views Section:**

Yes

**Compliance With Llm Reviewing Policy A Conservative:**

Affirmed.

**Discussion Potential:**

3

**Paper Summary:**

The authors argue that token logprobs from LMs should not be used as a basis for world modeling and instead see second-order-prediction as a viable alternative. The core of the argument is that there exist several biases that lead to a divergence of token and world event probability distributions. Such deviations might arise due to the misrepresentation of event occurrences in text datasets, as well as due to training and inference procedures that lead to a misalignment of the goals of language and world modeling. The argument is supported by an analysis of prompts and corresponding ideal output distributions, based on a formal framework. In addition, the paper identifies misconceptions that lead to wrong assumptions regarding the interpretation of token logprobs.

**Position:**

Yes

**Position In Title:**

Yes

**Related Work:**

2

**Strengths And Weaknesses:**

Strengths:
- The core claim that token logprobs should not be used as a basis for world modeling is well justified and supported by theoretical arguments from multiple perspectives throughout the paper.
- The paper is well structured. As a basis for the argument, the authors define central concepts and provide an overview of LM training and inference procedures which makes the paper very accessible.
- Section 6 uses explicit examples to demonstrate which misconceptions can occur in world modeling with LMs which makes the implications of the theoretical analysis tangible.

Weaknesses:
- As part of their position, the authors claim that second-order-prediction is the only theoretically sound way for LM-based world modeling. However, while the paper contains convincing arguments against using token logprobs for world modeling, there are comparatively few arguments that support second-order-prediction as a viable alternative. For example, the formal theoretical analysis in section 5 does not explicitly cover second-order-prediction as a use case.
- The paper could be strongly improved by adding empirical evidence that shows the potential of second-order-prediction, either by conducting experiments or by compiling results from prior work this topic in a more structured way.
- In section 5, the notation seems to be slightly incoherent. (E.g. in section 5.3: „If w^y states the agent always reports the outcome […]“. Was maybe w^x intended here? Further, in Equation 2, function f maps event x to event y (f(x)=y). However, in my understanding, f was introduced as a function mapping events to strings (see section 5.1). In general, section 5 might benefit from more clarity.

**Support:**

2

---

> ### Author Rebuttal · Authors · 2026-03-29
>
> We thank the reviewer for the careful and constructive feedback. We address the reviewer’s concerns and are happy to discuss these topics further.
>
> **Re. uniqueness and second-order prediction in the formal framework (W1):**\
> Given an input, neural language models have two types of outputs: the neuron or “hidden” states (in multiple layers) and decoded tokens or strings. Among hidden states, only the last layer, after softmax, is explicitly designed to represent probabilities. Decoded strings are open-ended, and, in particular, can also directly address probabilities (in various granularity levels). Our claim is that second-order prediction is the only theoretically sound method among these two options, which are the natural candidates since only they directly address probabilities without adaptation. We do not claim that no other conceivable method—e.g., putting hidden states or decoded strings through a function that gives probabilities—could exist. However, any other method must involve training or techniques that are not currently incorporated in neural language models. \
> We will clarify our position in the final version.
>
> **Regarding second-order prediction in Section 5 (W1) and the connection to Section 6 (Q1):**\
> The formal framework is indeed intended to show what assumptions are required for the different methods to be reliable, and, by extension, why those assumptions are generally not met when using logprobs. The case of second-order prediction corresponds to an agent possessing full knowledge of the intended distribution and intending to faithfully report this knowledge. Unlike the other cases, these properties are not misaligned with the source distribution or goals it implicitly contains.
> Although rather straightforward, we agree with the reviewer that his case should be addressed explicitly, and we add a short discussion of it.
>
> After establishing the necessary assumptions for each method, in §6 we analyze existing literature based on this understanding, identifying misconceptions that were seemingly overlooked. The connection is currently implicit: each pitfall can be traced back to a violated assumption in §5. We will add a brief bridging paragraph, making this structure explicit.
>
> **Re. deception and noise in second-order prediction (Q2) and empirical evidence (W2):**\
> We stress that our argument is not that second-order prediction is immune to practical difficulties, but rather that it does not suffer from the same theoretical mismatch as the other methods. In Case 2 (observed outcomes), biased reports in the training data are structurally embedded in the learned distribution. These biases reflect the learning objective when applied to the data. In Case 3 (second-order prediction), the same biases are not structurally required: reasoning about event probabilities can, in principle, override them through appropriate training and prompting.
>
> We present here a simple example: when asked to choose a random number, LLMs with standard instruction tuning almost always select 3 (in preliminary experiments with Llama-3-8B-instruct, we found this occurred in 100% of cases), yet the same model clearly states that the output should follow a uniform distribution. This dissociation — between what the model says and what it generates — illustrates exactly the gap between second-order prediction (saying the distribution) and response generation (acting on it), showing how second-order prediction can mitigate biases. Improving the model’s reasoning capabilities can extend this method to more complex scenarios.\
> We will briefly elaborate on this.
>
> **Re. notation in Section 5 (W3):**\
> Thank you for noting these. Specifically, in the definitions, a belief should possibly contain the input event only (e.g., for an unobserved outcome) and, in Eq.2, the indicator should use w^y instead of y (keeping the previously stated assumption of a single description per event). We will correct these in the final version.
>
> **Re. Yona et al. (2024) (Q3):**\
> Yona et al. run interesting experiments which align with our analysis. Our disagreement regards the interpretation of the results: they describe the gap between response decisiveness and generation probability as LMs being "poor at faithfully conveying uncertainty" (Abstract)—implicitly treating the generation probability as the ground truth for the model's internal confidence. They “aim to align the decisiveness … with the model’s intrinsic confidence” (Introduction), define a match as being “faithful” (§2), and conclude with the “desiderate that a language model’s generated response should reflect its intrinsic uncertainty” (Conclusion). In contrast, we argue that this gap represents misaligned tasks: decisiveness corresponds to target distribution estimation (through second-order prediction), and generation probabilities correspond to prediction. In other words, what Yona et al. identify as a weakness of models, we identify as a theoretically correct behavior.

---

> > ### Author Rebuttal · Reviewer_UPnH · 2026-04-08
> >
> > Thank you for the response. I still find that empirical evidence would strengthen the position but perhaps citing Yona et al. at could do as well.

---

### Official Review · Reviewer_oPZ9 · 2026-03-11

**Significance:** 4
**Argument Clarity:** 4
**Rating:** 4
**Confidence:** 4

**Questions:**

* How does explicitly prompting for a probability fundamentally overcome the base reporting bias to yield a true world distribution? Does this method not simply measure the model's structurally biased internal proxy of the world?
* Do large reasoning models offer more well calibrated second-order predictions than LLMs that have not been trained for CoT-style reasoning? I.e, if modulate the `enable_thinking` parameter in a hybrid reasoning model, can we empirically show that the reasoning model is better able to overcome its inherent biases?

**Alternative Views Section:**

Yes

**Compliance With Llm Reviewing Policy A Conservative:**

Affirmed.

**Discussion Potential:**

3

**Final Justification:**

Overall, the paper is thought provoking but its presentation could be improved. I have maintained my inital rating.

**Paper Summary:**

This paper highlights an apparently widespread misunderstanding of the significance of LM logprobs and specifically their interpretation as modelling the probabilities of the ground truth distribution rather than the generative output distribution. The authors clearly outline their position, including: the difference between distribution estimation and response prediction; how the various stages of LLM training map to different learned representations (probabilistic language modelling vs. maximization of expected reward), how modifying an input can lead to different interpretations of the logprobs output, and how confusion between these settings has led to claims in the literature which are unfounded.

**Position:**

Yes

**Position In Title:**

Yes

**Related Work:**

3

**Strengths And Weaknesses:**

## Strengths:
* Paper is well written and organized.
* The examples extracted from prior work are surprising and effective at expressing the timeliness and importance of expressing this position.
* Mapping between the various stages of LM training and what is actually being learned / rewarded is illuminating.
* The frequent use of illustrative examples effectively highlights the key conceptual arguments being made.

## Weaknesses:
* The proposed reliance on second-order prediction is still “highly-dependent on the quality of the model”. Indeed, the noted reporting bias in base pretrained LLMs will still be prevalent if the input is  structured as asking for explicit probabilities. Foundationally, this bias cannot be fully mitigated from any probabilistic world model.
* Second-order prediction relies on unstructured generation. When evaluating small LMs, logprobs may offer a more stable evaluation setting. However, I agree that regardless one should not misconstrue logprobs as reflecting the underlying ground truth distribution.
* Limited alternative views explored. While I agree that we cannot conclude that $\mathcal{p}_{lm}$ is unbiased with respect to the true world probabilities, we cannot make any theoretical guarantees when asking for second-order prediction either. The model may not be able to override such reporting biases in its training corpus despite side stepping the SFT mode collapse of allocating all probability mass to the maximal likelihood response.
* The practical recommendation to avoid interpreting LLM outputs as world probabilities and instead return to “fitting formal models” could be stressed earlier in the work. While second-order prediction may be the best way to elicit approximate event probabilities from an LLM, it is likely that they are ill suited to the task unless specifically post-trained for such behaviour.
* Limited empirical support for the proposed method. Since successful second-order prediction relies heavily on the models ability to reason and potentially override the priors learned from the pretraining corpus (i.e., bananas are green) it would be particularly compelling to show that large reasoning models offer sufficiently well calibrated world probabilities when prompted for second-order predictions compared to their non-reasoning counterparts. This could be demonstrated on hybrid reasoning models by repeating prompts with or without CoT reasoning enabled.

**Support:**

2

---

> ### Author Rebuttal · Authors · 2026-03-29
>
> We thank the reviewer for the detailed and perceptive feedback. We address the reviewer’s concerns and are happy to discuss these topics further.
>
> **Re. limitations of second-order prediction (W1-4; Q1):**\
> Our discussion addresses the LLM’s ideal behavior given the learning task it is presented with. In simple completion, the reporting bias reflects the optimal solution given the task definition; it is not a deficiency of the learning method,  but an expected consequence of distribution estimation on biased text. The same applies to mode collapse under instruction tuning. In second-order prediction, the optimal behavior—given an appropriate instruction and training for helpfulness—may be calibrated. This does not mean it is easy to achieve in practice, nor that other methods cannot be useful in many settings.\
> As a concrete example, in multiple-choice questions, models exhibit biases toward certain answer positions [1]. However, when asked to directly report the desired distribution, models will typically conclude that the prior should be uniform. So, for instance, when asked whether a coin landed on heads or tails with no additional information, reasoning-based second-order prediction should yield an unbiased answer. The bias in completion arises from the training data; the second-order prediction can, in principle, override it through explicit reasoning.
>
> **Re. empirical evidence for second-order prediction (W5):**\
> We mention multiple works that discuss various forms of second-order prediction and their performance (e.g., Tian et al., 2023; Kadavath et al., 2022; Yona et al., 2024). Our position is that these methods are *theoretically* preferred over methods that use logprobs: while the precise performance can vary depending on the data and reasoning capabilities, second-order prediction weaknesses are not fundamental to the method. We therefore call for researchers to focus on improvements in this direction.
>
> We also note that the banana color example illustrates how reporting bias affects completion only: the bias arises from the distribution of descriptions in the training data, not from the underlying event probability. A direct question about banana color has no inherent reason to exhibit the same bias under second-order prediction. On the contrary, discussions about the typical color of bananas, even in the training data, should not be biased towards green bananas.
>
> **Re. the effect of reasoning models (Q2):**\
> We believe that reasoning capabilities can help, as we suggest in the paper. A simple illustration: when asked to choose a number, LLMs with standard instruction tuning almost always select 3 (in preliminary experiments with Llama-3-8B-instruct, we found this occurred in 100% of cases), yet the same model clearly states that the output should follow a uniform distribution. This dissociation — between what the model says and what it generates — illustrates exactly the gap between second-order prediction (saying the distribution) and response generation (acting on it). Reasoning models, by making intermediate steps explicit, may be better positioned to close this gap, especially when more complex reasoning is required.
>
> [1] Pouya Pezeshkpour and Estevam Hruschka. 2024. Large Language Models Sensitivity to The Order of Options in Multiple-Choice Questions. In Findings of the Association for Computational Linguistics: NAACL 2024, pages 2006–2017, Mexico City, Mexico. Association for Computational Linguistics.

---

> > ### Author Rebuttal · Reviewer_oPZ9 · 2026-04-03
> >
> > > We mention multiple works that discuss various forms of second-order prediction and their performance (e.g., Tian et al., 2023; Kadavath et al., 2022; Yona et al., 2024).
> >
> > In the camera ready version, highlighting some of the prior *empirical* results would strengthen the position.
> >
> > > A direct question about banana color has no inherent reason to exhibit the same bias under second-order prediction. On the contrary, discussions about the typical color of bananas, even in the training data, should not be biased towards green bananas.
> >
> > I whole heartedly agree. My perspective is that leveraging second-order predictions simply moves the source of bias from the pretraining corpus to post-training. While we may reasonably conclude that LLMs will faithfully report that bananas are indeed yellow, I'm less confident of extrapolating that simple question to the general case. Sensitive topics such as politics etc. will likely be strongly biased by the post-training process. A more holistic discussion around the limitations of second-order predictions would impove the alternative views section.
> >
> > Overall, the paper is thought provoking but its presentation could be improved. I will maintain my rating.

---

### Official Review · Reviewer_eMvd · 2026-03-12

**Significance:** 3
**Argument Clarity:** 3
**Rating:** 4
**Confidence:** 2

**Questions:**

**Questions**
- Line 131, left column. For Bayes optimal classifier, it should be better to use [1] or [2] as references?
- Line 145, left column. This task has been defined as probabilistic supervised learning [3]. The term second-order prediction may not be appropriate for such a narrow case in this paper.
- Typos. Missing punctuations, line 252 left column, line 333 left column, line 361 right column.
- The proposed position is simply opposed to using softmax, in this way, the alternative view is simply using softmax. But the proposed position in the abstract states that second-order prediction is the only theoretically sound method. This is a claim about uniqueness, but the authors have not shown why it is unique. If there are other possible ways of representing event probabilities, uniqueness should not be claimed.
- Can you provide some more specific experiments to illustrate the gap between, for example, word and world distributions?


**References** \
[1] Sengijpta, Sailes K. "Fundamentals of statistical signal processing: Estimation theory." (1995): 465-466. \
[2] Tewari, Ambuj, and Peter L. Bartlett. "On the Consistency of Multiclass Classification Methods." Journal of Machine Learning Research 8, no. 5 (2007). \
[3] Gressmann, Frithjof, Franz J. Király, Bilal Mateen, and Harald Oberhauser. "Probabilistic supervised learning." arXiv preprint arXiv:1801.00753 (2018).

**Alternative Views Section:**

Yes

**Compliance With Llm Reviewing Policy A Conservative:**

Affirmed.

**Discussion Potential:**

2

**Final Justification:**

Based on the paper and rebuttal, the proposed position makes sense but the authors' argument and evidence support do not make it sound significant. Although the rebuttal has addressed my concerns, it does not change my evaluation because the manuscript in its current form does not change too much even if misunderstandings have been resolved. As a result, I vote for weak accept but am not very confident with the evaluation.

**Paper Summary:**

This paper proposes a position that world event probabilities obtained from language models should be reported by second-order prediction, in other words, direct prediction, instead of using token log probabilities. The authors describe distribution estimation and response prediction, showcase several ways of making inference with LMs, several ways of outputting distributions with LMs. Finally, the authors analyze the obstacles from word distribution to world event distribution. Alternative views and discussions are presented with practical takeaways.

**Position:**

Yes

**Position In Title:**

Yes

**Related Work:**

3

**Strengths And Weaknesses:**

**Strengths**
- The manuscript is well-written and easy to follow.
- The proposed position is of relevance and importance to the ICML community because LLMs are widely used and event probabilities are elicited in common scenarios.

**Weaknesses**
- There is lack of empirical results that support the claims.
- There are a couple of noteworthy typos to be fixed in order to meet publication quality.

**Support:**

2

---

> ### Author Rebuttal · Authors · 2026-03-29
>
> We thank the reviewer for the comments, corrections, and references, which we will incorporate into the final version.
>
> **Re. references for Bayes optimal classifier and probabilistic supervised learning (Q1-2):**\
> We thank the reviewer for these. We will incorporate the suggested references [1], [2] for the Bayes optimal classifier. We will mention “probabilistic supervised learning”, with reference [3]. We will still initially describe this method as “second-order prediction” to emphasize that the type of inference involves a higher level of abstraction.
>
> **Re. uniqueness (Q4):**\
> Given an input, neural language models have two types of outputs: the neuron or “hidden” states (in multiple layers) and decoded tokens or strings. Among hidden states, only the last layer, after softmax, is explicitly designed to represent probabilities. Decoded strings are open-ended, and, in particular, can also directly address probabilities (in various granularity levels). Our claim is that second-order prediction is the only theoretically sound method among these two options, which are the natural candidates since only they directly address probabilities without adaptation. We do not claim that no other conceivable method—e.g., putting hidden states or decoded strings through a function that gives probabilities—could exist. However, any other method must involve training or techniques that are not currently incorporated in neural language models. \
> We will clarify our position in the final version.
>
> **Re. empirical evidence (W1):**\
> Can the reviewer be more specific about what type of evidence is lacking?
> While our work is mainly theoretical, we do analyze (Section 6) many empirical works. Many works we mention find mismatches between distributions (e.g., Hu & Levy, Yona et. al., and many more in the appendix, such as works about calibration). Results that were interpreted as poor performance can be seen, given our analysis, as support of our position that the distributions are not identical.

---

> > ### Author Rebuttal · Reviewer_eMvd · 2026-04-01
> >
> > I have read the author rebuttal. Since my evaluation is already positive and I still lean towards acceptance but not very strong, I will not change my score.

---

### Official Review · Reviewer_ejMJ · 2026-03-13

**Significance:** 3
**Argument Clarity:** 1
**Rating:** 4
**Confidence:** 4

**Questions:**

I am a bit confused about the first case in Fig 1. In Fig 1, we expect the LLM to output a sequence of H/T. However, in Sec 2.1 (Distribution Estimation), we define the task of DE to estimate the probability of H/T.  How these two define the same task?

**Alternative Views Section:**

Yes

**Compliance With Llm Reviewing Policy A Conservative:**

Affirmed.

**Discussion Potential:**

2

**Final Justification:**

My question has been addressed, I hence raise my score to 4.

I think this position paper has value in an interesting problem, with insights that people might often misunderstand. Therefore, I am happy for acceptance.  I do not strongly support acceptance (not 5) as I am still not fully confident in the presentation.

**Paper Summary:**

This paper discusses the probability of words and worlds of the LLM generations. The paper argues that the different settings lead to istinct, potentially conflicting, desired output distributions. Particularly, for event probabilities, standard softmax may not be able to produce the desired probability. Instead, the paper argues that explicitly reporting the probability in the output, a method the paper describes as second-order prediction, is a better justified way.

**Position:**

Yes

**Position In Title:**

Yes

**Related Work:**

2

**Strengths And Weaknesses:**

Strength:

This paper discusses a very interesting and important topic: how to interpret LLM's token likelihood and how to obtain a more reliable probability for LLM's outputs. This topic is highly relevant to the ICML audience.  The discussion and reasoning in this paper are reasonable and look correct to me.

More importantly, in the Pitfalls in Computing World Probabilities section, the author discusses several common misunderstandings. This part shows the paper's position is not only correct, but also necessary, and can make practical impacts and inspire further discussions.

I really like the example of coin flipping, which provides a very intuitive approach to follow and understand this paper.


Weakness:

The major weakness of this paper is its structure. While the paper clearly states the position, the paper actually discusses more than this position.
The position argues that for world probabilities, people need to report the probability directly. However, besides this position, the paper also discusses other probability interpretations and other task types.  I think these tasks are equally important.
My suggestion is to modify the position to include the other cases. After that, the paper can explain each setup separately and call back to the position when discussing it.


Additionally, Section 2 seems to be distracting in this paper. I understand it's necessary when arguing different tasks, but how LLMs achieve the ability to complete the task does not seem to be a necessary preliminary knowledge. We can still do prompt-instructed generation even without the post-training stage --- it's just worse. Therefore, it might be better to assume ideal LLM and discuss how this is achieved in the appendix.  This is just my feeling; if the author or other reviewers disagree with this, it is totally fine and feel free to ignore this suggestion.


In general, I believe this paper discusses an important aspect of LLM and its distribution. However, the structure makes the paper harder to follow. I believe the paper can benefit from restructuring.
A potential idea might be: reorganize the position as I previously mentioned, and in the next section, discuss several tasks that we need probability from the model, which can give the reader a clear idea why we want to look at the probability and why this paper is relevant to their own research. Then, discuss how we will obtain the probabilities or responses, followed by their potential pros and cons.
But again, this is just my feeling and my own suggestion; if the paper has an underlying structure that I did not understand, I am also happy to further discuss.

**Support:**

2

---

> ### Author Rebuttal · Authors · 2026-03-29
>
> We thank the reviewer for their thoughtful comments and advice.
>
> **Re. the structure of the paper (W1-3):**\
> We appreciate the concrete suggestion for an alternative structure. We understand the logic in the reviewer’s suggestion, and we will explain the rationale for the current structure. We’d be happy to discuss this matter further.
>
> In our view (in the intro), the pitfalls and misconceptions we discuss are related to the transition in usage: traditional language modeling—as in Markov’s and Shannon’s work—is based on statistical modeling of language utterances, corresponding to *source distribution estimation*. This paradigm remains prominent in base-model training, relying on cross-entropy minimization. However, with the rising inference capabilities of LLMs, they have been increasingly applied to tasks in various domains, going beyond source distribution estimation.\
> Here, we identify two leaps that were not fully acknowledged. First, applying inference can create a *prediction* task, with a new, typically accuracy-based objective (commonly falling under classical machine learning). Second, the application to semantic questions falls under *target distribution estimation* instead of source distribution estimation. Both leaps contribute to the fact that output token probabilities may no longer represent the desired probability, despite the shared output space (e.g., probabilities over “heads” and “tails”).\
> We believe that a central reason for the lack of consideration of these transitions is the historical context: since language models already produce probabilities over the desired space, we absentmindedly assume they correspond to the intended distribution. For this reason, we chose to frame the paper through the evolution of LLMs.
>
> With this logic, we frame the paper as follows: as preliminaries (S2), we define the relevant tasks, showing the differences between them, in terms of objectives with respect to data. In S3, we analyze training and inference strategies, as they evolved in the field, in light of the tasks in S2. The analysis emphasizes how different steps are not merely practical techniques, but rather correspond to possibly different fundamental tasks. Importantly, S3 is part of an analysis and not merely background. Some details, such as post-training for instructions, even if not crucial for the main argument, amplify the transition in usage (i.e., the usage changed and training was adapted to this change). We emphasize, regarding the reviewer’s suggestion, that this analysis is all under the assumption of an ideal model—one that achieves its optimization goal. In S4, after establishing the fundamental differences between settings (under ideal conditions), we explain how they correspond to different practical use-cases. S5 is an optional, deeper analysis of how the different settings emerge when considering agents and their goals. We will clarify that this section can be skipped in a shallow reading. In S6, we demonstrate how these differences led to misconceptions. Although there are two main misconceptions involved, they both lead to unwarranted claims about the meaning of output probabilities as representing more than source distribution estimation.
>
> We agree that the motivation for each section should be made more explicit, and we will add brief orientation paragraphs in each section to help readers see how the pieces connect to the central position.
>
> **Re. the position (W1):**\
> Following the reasoning about the structure of the paper, we frame the positive position as the theoretically correct (in terms of compliance between the underlying task in the application) method to address probabilities when the application goes beyond text completion, which we broadly described as the “world”. Our main point is that, given the pitfalls of using logprobs, the remaining way to address probabilities is to reason about them. Therefore, the positive position is essentially equivalent to the additional negative claims, addressing the challenges of other approaches.
> We will restate and further elaborate on our position in the revised version.
>
> **Re. Fig. 1 Case 1 (Q1):**\
> In Case 1, the LM outputs a token (for simplicity, we assume a single token completion of either H or T), and we examine the next-token probability P(H | prompt). This probability is typically represented as logits (before Softmax). Assigning probabilities for tokens is exactly distribution estimation (of a conditional probability). When the model is used to generate a token, it applies a decision function as we mentioned in Section 2.3 (a process often described as “decoding”).
> The figure illustrates that, when the prompt describes a coin toss and then completes the utterance with an outcome, the (ideal) LM's token probability reflects the training data distribution rather than the true event distribution, due to reporting biases (e.g., a tendency to report “heads” more than “tails”). We will clarify this in the caption.

---

> > ### Author Rebuttal · Reviewer_ejMJ · 2026-04-02
> >
> > Thank you for the response. I would strongly recommend that the author provide the structure clearly in the revised version. This can help the reader follow the manuscript better.
> >
> > As my questions have been addressed, I will raise my score to 4. I do not strongly support acceptance (not 5) as I am still not fully confident in the presentation.

---

### Decision · Program_Chairs · 2026-04-30

**Decision:**

Accept (regular)

**Comment:**

The reviewers, contrary to myself, thought that it was necessary to point out the distinction between distribution estimation (what people say) and response prediction (what is true in the world).  The reviewers enjoyed the intuitive reporting bias token probability skew examples.  And while there was some disagreement around the phrasing that second-order prediction is the only theoretically sound method, the "warning" (reminder?) to the community was seen as necessary and timely.